# An Advanced Review: Polyurethane-Related Dressings for Skin Wound Repair

**DOI:** 10.3390/polym15214301

**Published:** 2023-11-01

**Authors:** Wenzi Liang, Na Ni, Yuxin Huang, Changmin Lin

**Affiliations:** Department of Histology and Embryology, Shantou University Medical College, Shantou 515041, China; 20wzliang1@stu.edu.cn (W.L.); nani@stu.edu.cn (N.N.); 21yxhuang@stu.edu.cn (Y.H.)

**Keywords:** polyurethane, wound dressing, natural polymers, synthetic polymers, composite material

## Abstract

The inability of wounds to heal effectively through normal repair has become a burden that seriously affects socio-economic development and human health. The therapy of acute and chronic skin wounds still poses great clinical difficulty due to the lack of suitable functional wound dressings. It has been found that dressings made of polyurethane exhibit excellent and diverse biological properties, but lack the functionality of clinical needs, and most dressings are unable to dynamically adapt to microenvironmental changes during the healing process at different stages of chronic wounds. Therefore, the development of multifunctional polyurethane composite materials has become a hot topic of research. This review describes the changes in physicochemical and biological properties caused by the incorporation of different polymers and fillers into polyurethane dressings and describes their applications in wound repair and regeneration. We listed several polymers, mainly including natural-based polymers (e.g., collagen, chitosan, and hyaluronic acid), synthetic-based polymers (e.g., polyethylene glycol, polyvinyl alcohol, and polyacrylamide), and some other active ingredients (e.g., LL37 peptide, platelet lysate, and exosomes). In addition to an introduction to the design and application of polyurethane-related dressings, we discuss the conversion and use of advanced functional dressings for applications, as well as future directions for development, providing reference for the development and new applications of novel polyurethane dressings.

## 1. Introduction

The skin is the body’s largest organ, and it is not only the first line of physiological defense, but also essential for survival. In addition, the skin has a complex self-regulatory function [1]. In response to harmful stress, such as pathogens, thermal, mechanical, and chemical hazards, the skin responds to regulate local and systemic homeostasis. The structure of the skin consists of three layers: superficial epidermis, deeper dermis, and subcutaneous hypodermis. The epidermis is mainly composed of keratinocytes and undergoes constant renewal where basal epidermal stem cells with high proliferation potential produce new daughter cells or translocation expansion cells [2]. Skin can regulate water and permeate oxygen and carbon dioxide, and its sensory properties can affect thermoregulation and immune function [3]. Wound healing refers to a series of physiological processes in which damaged tissues are repaired through various cells and interstitial tissue after skin injury (Figure 1) [4], and it mainly occurs in skin tissues after traumatic injury, infectious ulcers, or burns. Rapid wound healing and rapid regeneration of damaged skin are essential to restore barrier function. Chronic wounds are also known clinically as hard-to-heal wounds because they are more difficult to heal and take a long time to treat, such as in people with diabetes or those who are chronically bedridden [5]. Chronic wounds occur when the normal healing process stalls, which can seriously affect patient quality of life and place a heavy burden on healthcare systems [6].

Recovery of skin wounds requires optimal temperature, humidity, pH, and oxygen. A moist environment and appropriate pH can maintain the activity of cells and enzymes, which is conducive to wound healing, as well as resistance to infection and protection from harmful external factors. As a result of a moist wound environment, autolysis debridement occurs, pain is reduced, scarring is reduced, collagen deposition occurs, blood vessels form, and migration of keratinocytes is enhanced [7]. To achieve an ideal wound environment, dressings may be selectively applied outside the wound to promote repair or to reduce the risk of infection and relieve or reduce pain. Local treatment requires choosing the right wound dressing based on the wound’s characteristics, location, depth, area, level of exudation, presence of infection, stage of healing, and skin type [8]. The ideal dressing should fulfill the following requirements (Figure 2): ensure physical continuity of the wound; have ideal fluid handling capacity; have good antimicrobial activity against a wide range of bacteria and fungi; provide optimal thermoregulation, humidity, and pH; exhibit good cytocompatibility; support growth and proliferation of fibroblasts; protect against rejection (inhibit granulation, fibronectin formation); be non-allergenic, be comfortable to use, permeable to oxygen and carbon dioxide, and cost-effective; do no harm to the ulcer edges [8,9,10,11]. A number of different types of dressings, including polyurethane foam, membrane dressings, hydrocolloid dressings, hydrogel dressings, alginate dressings, semi-permeable polyurethane membrane dressings, and genetically engineered flax dressings, are used all over the world [8,12]. Doctors will choose the appropriate dressings according to medical knowledge, treatment experience, clinical characteristics of wounds, and the specific needs of patients.

Polyurethane is an important biomolecular material, which has been the focus of research, and it plays a vital role in the field of artificial organs, medical devices, and medical materials. The reason for choosing polyurethane as the substrate material for dressings is mainly based on the following reasons: (1) Polyurethane materials are composed of soft and hard segments, and the performance of polyurethane materials can be adjusted by changing the type and proportion of soft and hard segments; (2) Polyurethane material has excellent mechanical properties, and it is easy to be processed; (3) Polyurethane materials have excellent biocompatibility and low toxicity. For the synthesis of polyurethane, it is a kind of polymer material containing carbamate group (-NH-COO-), the main synthesis method is by polyether, polyester, or polycarbonate diols and diisocyanate for addition reaction, and then by chain extenders to expand the chain into polymer. The main chain of polyurethane is composed of soft and hard segments. Due to the thermodynamic incompatibility between the soft and hard segments, the performance of polyurethane is related to the chemical structure and proportion of the soft and hard segments, which further affects the performance of polyurethane dressing materials [13,14,15,16,17,18]. Conventional textile fiber wound dressings usually become infiltrated with wound secretions and newly formed soft tissue. The wound secretions and newly formed soft tissue infiltrate are difficult to remove, often resulting in secondary skin damage. In the early days, modern dressings composed of polyurethane polymers were reported to be more effective, comfortable, convenient, and economical compared to other traditional dressings. The advantages of polyurethane dressings have played an important role in outpatient settings and inpatient care [13,14,15,16,17,18]. However, drawbacks such as the inability to control leakage, increased cost of care, and poor cost-effectiveness of polyurethane polymer-related dressings have also been reported [16,19,20]. Although polyurethane-related dressings have been commercialized, there are still many functional deficiencies [21,22]. Currently, polyurethane dressings have been further improved to facilitate wound healing in the early stages and to reduce patient pain and discomfort, achieving the goal of minimizing wound healing time and improving cost-effectiveness [21,23]. The most common approaches to the development of innovative and improved polyurethane wound dressings include the synthesis and modification of biocompatible materials to improve biomedical performance, to overcome undesirable biological functions of polyurethane polymers, antimicrobial functions, and to impart mechanical and thermal properties of biomolecules. In addition to delivering unique and versatile functionality, these new polyurethane polymeric materials also perform specific biochemical functions, making them as the ideal wound dressings [24].

Up to now, many investigators have published reports on the effects of modified polyurethane polymer dressings on wound healing, but few of them have analyzed research in the field. The purpose of this work is to analyze the research results of polyurethane-related skin wound dressings, and to provide support for further research in related fields by comprehensively sorting out and analyzing the unsolved problems.

## 2. Methods

### 2.1. Search Strategies

The literature search was conducted using PubMed and Web of Science. Articles published with “polyurethane”, “dressings”, and “wound” as titles or abstracts from 2019 January to 2023 October were retrieved. The search equation used for PubMed was ((polyurethane [Title/Abstract]) AND ((dressings [Title/Abstract]) OR (wound [Title/Abstract]))). The search equation used for Web of Science was (TS = (polyurethane) AND (TS = (dressings) OR TS = (wound))).

Study inclusion criteria included (1) studies of polyurethane composite dressing synthesis methods and characterization, (2) studies on biocompatibility and medical performance assessment of dressings, and (3) studies involving natural polymers, synthetic polymers, or other bioactive ingredients.

Study exclusion criteria included (1) studies that did not involve research on wound dressings, such as studies on e-skin composed of polyurethane, (2) reviews and meta-analyses, (3) papers that were not in English, (4) papers that were not in the category of empirical research, (5) papers that had outdated ideas or repetitive arguments.

### 2.2. Search Results

A total of 125 articles were retrieved from PubMed. A total of 757 articles were retrieved from Web of science. Finally, 45 reports were selected based on the inclusion/exclusion criteria and were included for data extraction, as described in Section 3.1, Section 3.2 and Section 3.3.

### 2.3. Categorization and Display Strategies

Polyurethane composites are widely used as dressings to treat skin wounds. With additives that provide biocompatibility, polyurethane wound dressings can be functionally controlled. Polyurethane is usually combined with a polymer to form a conforming material, and these polymers are generally divided into natural polymers and synthetic polymers [11,25]. A natural polymer is composed of biomolecules derived from nature, such as microorganisms, animals, or plants that can mimic the original cellular environment and extracellular matrix very closely. The main natural polymers used in synthetic dressings include silk protein, pullulan, chitosan, cellulose, alginate, glucan, collagen, elastin, carrageenan, pectin, agarose, hyaluronic acid, fibrin, chitin, and gelatin [26,27]. Synthetic polymers comprising the dressing include polyglycolic acid, polyethylpyrodanone, polylactic acid, polyhydroxyethyl methacrylate, polycaprolactone, polyvinyl alcohol, and polylactic acid-co-glycolic acid [28]. Furthermore, other bioactive ingredients, such as essential oils, dextrans, cells, acellular matrices, propolis, vitamins, growth factors, thyroid hormones, proteins, insulin growth factors, enzymes, and nanoparticles that fight bacteria, are also used in the synthesis of wound complex dressings [27,29].

## 3. Results

### 3.1. Natural-Based Polymeric Wound Polyurethane Dressing

Many natural materials, usually including collagen, chitosan, hyaluronic acid, vegetable oil, tannic acid, thymol, lignin, and some animal sources of skin, have been successfully used in the production of polyurethane dressings [26,27].

#### 3.1.1. Collagen

Collagen dressing is highly absorbent, can control wound exudation, and protect the wound. Blending collagen products with polyurethane materials can effectively improve the performance of collagen dressings [30]. Composite hydrogels based on collagen crosslinked with polyurethane and metal-organic frameworks (MOFs) with aluminum as metallic center were synthesized by the microemulsion method. It was found that the entanglement of polyurethane, collagen, and MOFs was made by hydrogen and coordination bonds promoted by the chemical structure of the MOF, leading to a semi-crystalline rough surface with interconnected porosity and aggregates of round shape, enhancing the thermal degradation resistance, mechanical degradation resistance and biocompatibility [31].

#### 3.1.2. Chitosan

Chitosan is a linear polysaccharide with acetyl and amine group branches, and in acidic conditions the amino group is converted to a polycation (type IV amine). It is a biocompatible, non-toxic, and biodegradable biopolymer that can be used in the manufacture of a wide range of medical materials [32]. Studies have confirmed the benefit of adding the chitosan to polyurethane (PU) polymers and the PU/chitosan scaffolds developed by electrostatic spinning, which allows for the formation of a homogeneous structure in the scaffold fibers. The effect of PU/chitosan on the morphology and cellular activity of fibroblasts was assessed and it was found that this scaffold became a more favorable environment for fibroblast survival and growth. This suggests that PU/chitosan dressing can be a potential wound dressing. In addition, the study proposes another beneficial polysaccharide, hyaluronic acid [33], as described in Section 3.1.3.

#### 3.1.3. Hyaluronic Acid (HA)

Hyaluronic acid, a non-sulfated glycosaminoglycan, is a major component of the skin’s extracellular matrix and is involved in the processes of angiogenesis, inflammatory response, and tissue regeneration. Due to its excellent biocompatibility, biodegradability and hydrophilicity, HA has been widely used in the synthesis of wound dressings [34]. The preparation of PU/St (starch) and PU/St/HA core-shell nanofibers was accomplished by electrostatic spinning. To evaluate the properties of PU/St nanofibers and PU/St/HA nanofibers in vitro, mouse fibroblasts were used. For the purpose of evaluating cell survival and proliferation, the MTT assay was employed. It was found that PU/St/HA core-shell scaffolds did not significantly alter cell survival and proliferation, and that they were more biocompatible and did not cause cytotoxicity. In vivo studies in rats have shown that core-shell PU/St/HA wound dressings keep the skin moist, do not produce excessive wound exudate, have a higher quality of tissue repair, and confer faster wound healing [35]. At −20 °C, a dihydrazide-modified waterborne biodegradable polyurethane emulsion (PU-ADH) and oxidized hyaluronic acid (OHA) were autonomously crosslinked to form a hybrid hyaluronic acid−PU (HA-PU) cryogel by hydrazone bonding. Through the specific macroporous structure (~220 μm) formed by the polymerized PU-ADH particles and long-chain OHA, the dried cryogel swelled rapidly within minutes and could absorb blood or water up to 16 and 22 times its dry weight. This instantaneous shape recovery capability facilitated rapid hemostasis in minimally invasive procedures. In addition, the cryogel had a greater biocompatibility than gauze, enhanced blood coagulation, and activated endogenous coagulation after about 2 min of use. Using the same composition as HA-PU low-temperature gels, injectable HA-PU hydrogels with good self-healing properties were prepared at room temperature. In vivo evaluations of animals demonstrated that the cryogel was extremely effective in rapid wound healing, reduced immune-inflammation, and promoted angiogenesis and regeneration of hair follicles [36].

#### 3.1.4. Vegetable Oil

Vegetable oils are one of the most important biomass raw materials for synthesizing polymers. The main components of vegetable oils are triglycerides. In addition, there are also some highly reactive active sites in vegetable oil, including double bonds, hydroxyl groups, and ester groups, which provide the possibility for the preparation of various polymers with different structures and functions. Due to the wide source, renewable, non-toxic, and biocompatibility characteristics, vegetable oil-based polymers are widely used in the production of biomedical materials [37]. A novel soybean oil polyol with built-in urethane and quaternary ammonium groups was synthesized by a non-isocyanate route using environmentally friendly and renewable carbonated soybean oil as raw material. Polyurethane wound dressing was prepared by the reaction of isophorone diisocyanate, castor oil, and this polyol. Antimicrobial activity and cytocompatibility of the dressing were good. Depending on the hydration state and the dry state, the dressing can have a tensile strength of 5 MPa and 17 MPa. The equilibrium rate of water absorption was 50%, and the water vapor transmission rate was 390 g per square meter per day. Evaluation of an optimized dressing on a full-layer unsterilized wound showed that wound healing progresses well, with the regenerated skin achieving a tensile strength of about 80% of normal healthy skin at day 21 post-injury [38].

#### 3.1.5. Tannic Acid (TA)

TA is a naturally occurring plant-derived polyphenol. TA can be used as a crosslinking agent by supramolecular or physicochemical methods and is widely used in the production of skin adhesives and wound dressings [39]. Biologically active biomimetic skin hydrogel band-aids could be developed utilizing imidazolidinyl urea-based reinforced polyurethane (PMI) combined with TA (TAP hydrogels). As a result of multiple non-covalent interactions between TA and the PMI hydrogel network, the mechanical properties of the TAP hydrogel were strengthened, and the wound dressing’s structural integrity could be maintained under local stresses. For its excellent self-recovery and anti-fatigue properties, TAP30% hydrogel provides excellent comfort. Owing to its good anti-moisture, adhesive, organ hemostasis, excellent anti-inflammatory, antioxidant properties, and antimicrobial activity, when diabetic mice were treated with TAP hydrogel, they were able to recover from skin incisions and defects more quickly. The therapeutic efficacy of TAP hydrogel was further investigated and shown to be effective in a diabetic mouse model infected with *Staphylococcus aureus* [40]. A study has shown a water retaining separable adhesive hydrogel wound dressing composed of TA. The incorporation of TA with abundant catechol moieties provided the hydrogel with improved mechanical properties, good tissue adhesion, and hemostatic ability. Then, a hydrophobic polyurethane-related coating was encapsulated on the surface of the hydrogel to maintain a high water content of the hydrogel for a long time [41].

#### 3.1.6. Thymol

A versatile portable electrostatic spinning device has been created, featuring an adjustable perfusion rate and a high voltage capacity of up to 11 kV. Thymol, a natural antimicrobial compound, was doped into ethanol-soluble polyurethane (EPU) skin-like W&B nanofibrous membranes to give them antimicrobial activity. EPU-like skin-type waterproof and breathable nanofiber membranes with antimicrobial activity were prepared using a customized device. Excellent uniformity of structure was observed in the final nanofibrous membrane, which is composed of fluorinated polyurethane (FPU), EPU, and thymol. The membrane has a tensile stress of 1.83 MPa, and a tensile strain of 453%. The permeability is 3.56 kg m^−2^ d^−1^, hydrostatic pressure is 17.6 cm H_2_O, and antimicrobial activity is high [42].

#### 3.1.7. Lignin

In the pathophysiology of wounds, a number of external and internal factors contribute to impaired wound healing. In particular, oxidative stress is an important factor in inhibiting wound healing [43]. There is uncertainty about the biocompatibility of lignin, which is a plant-derived antioxidant, and it remains underdeveloped as a biomaterial limiting its biomedical applications [44]. A lignin nanogel has been developed and explored for its therapeutic effects in skin wounds. Lignin derived from coconut shells shows good antioxidant properties. In a thermosensitive nanogel based on polyethylene glycol (PEG), polypropylene glycol (PPG), and polydimethylsiloxane (PDMS) polyurethane copolymers, lignin was incorporated. No significant cytotoxicity was observed with nanogels containing doped lignin. As the result of lignin nanogel antioxidant properties, oxidative stress-induced apoptosis in LO2 cells was prevented. In a mouse burn wound model, lignin nanogels accelerated wound healing, a result further supported by immunostaining for the cell proliferation marker Ki67. In this regard, lignin nanogels may prove useful as a wound dressing that promotes wound healing through its antioxidant properties [45]. For wound dressings, porous nanocomposite polyurethane foams that contain nanolignin (NL) and coated with natural antimicrobial propolis had also been reported. The foams were soaked in ethanol extract of propolis (EEP) after synthesis. NL and EEP coatings improved the foams’ hydrophilicity as measured by contact angle. Furthermore, the EEP coating enabled the dressing to have significant antimicrobial effect and good cytocompatibility. The effectiveness of PU-NL/EEP on wound healing has also been demonstrated in a rat whole skin wound model [46].

In addition, one-step foaming methods have also been proposed for lignin-based polyurethane foams (LPUFs), where fully biobased polyether polyols partially replace petroleum-based feedstocks. LPUF skeletons contain trace amounts of phenolic hydroxyl groups (~4 mmol) that act as a direct reducing and capping agent for silver ions (<0.3 mmol). A lignin composite foam has been developed with improved mechanically and thermally properties [47].

#### 3.1.8. Peppermint Extract

Wound dressings containing herbal extracts with high antimicrobial properties and a nanoscale-controlled release system have been shown to facilitate the healing of ulcerated wounds [48,49]. Herbal extracts of peppermint have been used to treat bacteria and inflammation [50], and a novel mint extract added to polyurethane-based nanofibers has been shown to be useful for diabetic wound healing. In order to optimize the release of the extracts, gelatin nanoparticles (CGN) have been crosslinked with the extracts and ultimately incorporated into nanofibers. Direct incorporation of extracts into a polyurethane matrix also controlled extract release. With an antimicrobial rate of 99.9%, the wound dressing was able to absorb *Staphylococcus aureus* and *Escherichia coli*. The in vivo study found that this extract improved wound healing after using this extract as an active compound. Inflammation is significantly reduced in wounds treated with nanofiber extracts, according to histopathological studies. In addition, skin of treated individuals had characteristics more similar to normal skin, including the epidermis exhibited thinning, the reticular ridges appeared normal, and the appendages grew back [51].

#### 3.1.9. Gelatin

Gelatin is one of the most biodegradable and biocompatible polymers derived from the hydrolysis of collagen. It helps in cell adhesion and speeds up the healing process of wounds, but it is prone to degradation [52]. Blending of natural and synthetic polymers could improve structural stability. Introducing 20% PU to gelatin scaffolds (Gel80−PU20) results in a significant increase in the degradation resistance, yield strength, and elongation of these scaffolds without altering the cell viability. In vivo studies using a mouse excisional wound biopsy grafted with the scaffolds reveals that the Gel80−PU20 scaffold enables greater cell infiltration than clinically established matrices [53]. Personalized medicine is made possible by three-dimensional (3D) printing of soft biomaterials. By developing different forms of 3D-printed biomaterials, artificial organ fabrication can be facilitated and desired properties can be incorporated into biomaterials. In order to develop 3D-printable gelatin methacryloyl (GelMA) polyurethane biodegradable hydrogels and cryogels, GelMA was combined with dialdehyde-functionalized polyurethane (DFPU). A 3D-printed biomaterial with high print resolution, smart functionality, and biocompatibility was presented by the GelMA-PU system, demonstrating a combination of self-healing and 3D-printing capabilities. With GelMA-PU, the ink pool for biomaterial 3D printing has been expanded, allowing applications such as tissue-engineered scaffolds, minimally invasive surgical instruments, and electronic wound dressings [54]. It has also been shown that an absorbable gelatin sponge combined with a polyurethane film could be effectively used for skin reconstruction of bone or tendon exposed wounds [55].

#### 3.1.10. Dextran

Dextran is a polysaccharide with good biocompatibility and degradability and an interfering effect on coagulation and hemostasis, which can be used to compensate for the adverse effects of antimicrobial agents in wound dressings. It has great potential for application in medical materials and tissue engineering [56]. To develop antimicrobial wound dressings, pH-stimulated drug release nanofiber membranes of polyurethane/dextran were developed. Dextran was added to polyurethane to increase hydrophilicity, air permeability, percent adsorption value, and biodegradability. Dextran can also be used as a reinforcing filler in polyurethane matrices. Dextran induces high platelet adhesion and hemostasis, which is essential for promoting the wound healing process. In addition, 20 wt% dextran-loaded membrane (PU/20D) enhanced cell proliferation, attachment, and survival of fibroblasts [57]. It has been shown that polyurethane prepolymers could be made into wound dressings by sol-gel hydrolysis polycondensation reaction and surface modification by dextran. The biological properties of the final dressings were improved, and the dextran anhydride modification resulted in dressings with low hemolysis rates and prolonged clot formation [58].

In summary, the advantages of natural polymers as a source of dressings are that they are widely available, renewable, degradable, non-toxic, and biocompatible. The disadvantages are the complex structure of natural polymers, complicated extraction process, and poor mechanical properties. The chemical structure of natural-based materials is shown in Table 1, and the content of the natural-based polyurethane materials discussed above are shown in Table 2.

### 3.2. Synthetic Polymer and Inorganic Modified Polyurethane Dressings

Synthetic polymers are chemically synthesized in the laboratory and are also known as artificial polymers. In order to improve the biological properties of such polymers to reach their potential as wound dressings, various surface and bulk modifications have been applied [28].

#### 3.2.1. Povidone-Iodine (PVP-I)

Povidone-iodine has potent  broad-spectrum  activity  against  bacteria, virus, fungus, and protozoa [59]. PVP-I polyurethane dressing (Betafoam) is a new type of polyurethane dressing impregnated with 3% of PVP-I [60]. For the first time, the effect of PVP-I dressings on split-thickness skin graft donor area wounds was validated in a clinical case. The efficacy and safety of PVP-I dressing were compared with that of vanilla oil gauze and cellular water dressing. This was primarily determined by observing the degree of donor site epithelialization. PVP-I dressing provided better wound healing with significantly shorter time to complete epithelialization (approximately 14 days). PVP-I foam dressing allowed for easier wound care, less bleeding and easier removal of dressing adhesion, and better exudate management. It offers significant clinical advantages and is cost-effective [61]. Betafoam has been verified to be effective in wound healing in a rat skin healing model, showing good performance in re-epithelialization, angiogenesis, collagen deposition, and tissue invasion [62].

#### 3.2.2. Polyacrylamide (PAAm)

In tissue engineering, drug delivery, smart materials, and drug delivery systems, PAAm is a polymer used extensively for its excellent mechanical properties, hydrophilicity, and biocompatibility [63]. Waterborne polyurethanes are able to provide many functional groups that make polyurethanes easy to functionalize when interacting with other chemicals [6,64]. After rapid-curing by UV, a mechanically flexible PU-PAAm hydrogel skin dressing with good adhesion was developed. The polyurethane component of the PU-PAAm hydrogel acts as a “bridge”, accelerating the interpenetrating polymer network (IPN) formation, which consists of a physically crosslinked polyurethane network surrounded by a chemically crosslinked PAAm network. Due to its unique IPN structure, the hydrogel is exceptionally stretchable and ductile. During application, the hydrogel and skin form hydrogen bonds and electrostatic interactions, which ensured strong adhesion, and the dressing is applied without irritating the skin and causing skin damage. L929 fibroblast experiments were used to validate the biocompatibility of PU-PAAm hydrogel, and rabbit skin wound healing experiments further confirmed the remarkable skin regeneration-stimulating ability of PU-PAAm hydrogel [6]. Using super-tough thermoplastic polyurethane (HTPU) hydrogel and chemically crosslinked PAAm as the first and second network, HTPU/PAAm double-network hydrogels were synthesized by a one-step radical polymerization in a study. The toughness and strength of this polyurethane-related hydrogel were greatly improved, and it has broad application prospects in wound dressing [65].

#### 3.2.3. Polycaprolactone (PCL)

Polycaprolactone is an important polymer with good mechanical properties, miscibility with other polymers, and biodegradability [66]. PCL/Gel scaffolds have shown significant value in skin tissue engineering. However, these scaffolds have poor antimicrobial properties and are unsuitable for water vapor transmission [67,68]. PCL/Gel scaffolds are electrostatically spun on a dense membrane consisting of polyurethane/ethanolic extract of propolis (PU/EEP). As an upper layer, PU/EEP membranes protect the wound from external contamination and dehydration, and the PCL/Gel scaffolds act as a lower layer to promote cell proliferation and adhesion. Antimicrobial assays showed significant antibacterial activity against *Staphylococcus aureus*, *Escherichia coli*, and Staphylococcus epidermidis. The PU/EEP-PCL/Gel bilayer dressing had high hydrophilicity, biocompatibility, and biodegradability. In vivo experiments demonstrated that the double-layer wound dressing significantly promoted skin wound healing and collagen deposition in Wistar rats [68]. A two-layer wound dressing has been prepared using an electrostatically spun PCL/CS fiber mat as the inner layer, and polyurethane foam-coated EEP as the top layer. An electrostatically spun mat consisting of uniform nanofibers with enhanced hydrophilicity, swelling rate, and degradation properties is prepared by mixing PCL and CS solutions [69].

#### 3.2.4. Polylactic Acid (PLA)

Polylactic acid (PLA) is biodegradable and biocompatible, and it is a polymer widely used in biomedical materials. However, the brittleness and weak mechanical properties of polylactic acid nanofibers limit their application. PU has excellent elasticity and mechanical properties suitable for specific tissues. When PLA and PU are used together, in addition to improving the mechanical properties of wound dressings, they can also promote biodegradation [70]. When PLA is added to wound polyurethane dressings (PU/PLA, 50/50, *w*/*w*), the dressings absorb wound exudates, dry quickly, are comfortable, and have high biocompatibility to support fibroblast growth [71,72]. In a study, novel hollow nanofiber materials were produced by the coaxial electrospinning method from PU/PLA blend nanofibers of different weight ratios (20:80, 40:60, 50:50, 60:40, and 80:20). Moreover, hollow PU/PLA nanofibers were observed to be 2–4 times thinner than solid PU/PLA nanofibers. The production of hollow nanofibers in the range of 235–518 nm was achieved. It was determined that the biomedical material, which has the highest liquid absorption capacity with a value of 756% and can dry in 10 min, is PU/PLA (50/50, *w*/*w*) nanofiber. [72]. In addition, a prospective, comparative, randomized clinical study showed that polylactic acid membrane could improve the prognosis of cracked skin graft donors [73].

#### 3.2.5. Polyethylene Glycol (PEG)

PEG is formed by the stepwise addition polymerization of ethylene oxide with water or glycol. Polyethylene glycol polymer, because of its good water solubility and good compatibility with many organic components, has been widely used. Chen et al. prepared a PEG and triethoxysilane (APTES) modified high absorption polyurethane foam dressing (PUESi) [74]. The research results indicated that PUESi dressings not only had good adhesive resistance and deformation absorption ability, but also could effectively promote wound healing. Pahlevanneshan et al. synthesized polyurethane foam with polyethylene glycol, glycerin, nano-lignin (NL), 1, 6 diisocyanate hexane and water as foaming agents, and soaked it with propolis ethanol extract (EEP) [46]. The results indicated that PU-NL/EEP material had high cell viability and cell adhesion, and in vivo wound healing experiments were conducted using a Wistar rats’ full-thickness skin wound model, confirming that PU-NL/EEP material exhibited high wound healing effects. In addition, Vakil’s team developed a polyurethane-based polyethylene glycol hydrogel with cell compatible shape memory function, and then physically mixed plant-based phenolic acid onto the hydrogel scaffold so that it could be easily transported to the wound site, thereby increasing the wound healing efficiency and reducing the risk of infection [75].

#### 3.2.6. Polyvinyl Alcohol (PVA)

PVA is a white, stable, and non-toxic water-soluble polymer made from vinyl acetate through polymerization and alcoholysis. It is an extremely safe organic polymer with non-toxic and good biocompatibility, widely used in wound dressings and artificial joints. Hussein’s team used dual spinneret electrospinning technology to prepare polyurethane and polyvinyl alcohol gelatin (PVA/Gel) nanofiber scaffolds [76]. By adding cinnamon essential oil (CEO), the inhibitory effect of loaded low-dose nanoceria PU/PVA Gel NFs on Staphylococcus aureus was improved, and the therapeutic effect on diabetes wounds was effectively improved. Carayon’s team prepared multiple types of polyurethane-polylactic acid-polyvinyl alcohol composite porous matrices (CPMs), and the research results showed that the average porosity of CPMs was 69–81%, making their pore size more suitable for skin regeneration [77].

#### 3.2.7. Tributylammonium Alginate Surface-Modified Cationic Polyurethane (CPU)

Disintegration of membranes and death of bacteria can be caused by positively charged cationic polymers. Low cytotoxicity and long-lasting antibacterial activity are the main advantages of cationic polymers. A transparent tri-butylammonium alginate CPU skin dressing has been created for use in the treatment of full-thickness wounds. The surface-modified polyurethane of this dressing has improved hydrophilicity and tensile Young’s modulus between 1.5–3 MPa, which is close to natural skin. MTS and scratch assays were used to assess cell viability and showed that the dressings were cytocompatible and could promote fibroblast migration. Surface-modified CPU polymers are highly inhibitory to Gram-positive *Staphylococcus aureus* and Gram-negative *Escherichia coli* bacteria. In vivo experiments in rats showed that surface-modified CPU dressings promote the rate of wound healing, shorten the period of persistent inflammation, enhance collagen deposition, and promote blood vessel formation [78].

#### 3.2.8. Cellulose Acetate/Polyurethane Nanofibrous Mats Containing Reduced Graphene Oxide/Silver Nanocomposites and Curcumin

Nanofiber scaffolds can be prepared by electrostatic spinning using polyurethane and cellulose acetate as raw materials. Reduced graphene oxide/silver nanocomposites have strong antimicrobial activity. In order to prevent the aggregation of silver nanoparticles (AgNPs), AgNPs were loaded onto reduced graphene oxide and nanocomposites were prepared using a green and convenient hydrothermal method. The electrostatic spinning method was used to prepare scaffold materials containing reduced graphene oxide/silver nanocomposites, curcumin, or both. The MTT cell proliferation assay showed that the scaffold has good biocompatibility. Evaluation of antimicrobial activity showed that the scaffold is able to inhibit both Gram-negative and Gram-positive bacteria. In vivo experiments and histopathological studies showed that scaffolds containing graphene oxide/silver nanocomposites and curcumin can promote the rate of skin wound healing, suggesting that nanomaterials have a good biomedical potential for wound healing [79].

#### 3.2.9. Nanosized Copper-Based Metal-Organic Framework

Nano-Cu-BTC (copper (II)-benzene-1, 3, 5-tricarboxylate) is doped into polyurethane foams (PUF), through a polyaddition reaction of castor oil and chitosan with toluene 2, 4-diisocyanate, to improve the functionality of the dressing by modifying the PUF surface. The physical and thermal properties of Nano-Cu-BTC-PUF (PUF@Cu-BTC) were compared with those of control PUF, including swelling rate, phase transition, thermal gravity loss, and cytocompatibility, and they were evaluated for inhibitory activity against methicillin-resistant *Staphylococcus aureus*, Pseudomonas aeruginosa, and Klebsiella pneumoniae. The antimicrobial activity of PUF@Cu-BTC against the tested bacteria is significant and selective, and its cytotoxicity toward mouse embryonic fibroblasts is low. According to Cu (II) ion release assay, PUF@Cu-BTC is stable for 24 h in phosphate-buffered saline. Because PUF@Cu-BTC displays selective bactericidal activity and low cytotoxicity, it has potential for use as a skin wound dressing [80].

#### 3.2.10. Silver

Silver has strong antimicrobial activity and is commonly used in wound care. However, silver also has the potential to have toxic effects on skin cells, which can affect wound healing. There is some evidence that short-term use of dressings containing nanosilver is feasible in infected wounds, but the use of silver-containing dressings on clean wounds and closed surgical incisions is not appropriate. Ideal silver preparations are silver nanoparticles (AgNPs) and silver-coated polyurethane dressings for negative pressure wound therapy [81]. A new method of incorporating AgNPs onto the surface of polyurethane nanofibers has been proposed. Before the electrospinning process, AgNO_3_ and tannic acid were added to PU solution to make the AgNPs uniformly distributed on the surface of PU nanoparticles [82]. Thiol-terminated polyurethane prepolymers with two different molecular weight PEGs and terminated propargyl polyurethane crosslinkers have been prepared and polymerization reactions carried out with and without the addition of silver salts. A radical-mediated step-growth polymerization reaction and resultant thioether linkages created during polymerization lead to high conversion of the starting macromonomer and the formation of a hydrophilic network. Even in the hydrated state, the materials offer desirable dimensional strength and flexibility, as evidenced by its high tensile strength, good extensibility, and minimal permanent set. The reduction in silver salt during network formation both from reaction with free radicals and residual DMF solvent available in the reaction medium led to the formation of AgNPs. This dressing showed little toxicity to fibroblasts, high bactericidal and fungicidal activity, and good biocompatibility. No significant reduction in cell migration was observed with AgNPs dressings [83]. In addition to antibacterial properties, studies have also shown that the addition of AgNPs can improve mechanical properties (tensile strength and elongation at break) [71,84].

In summary, the advantages of synthetic polymers as a source of dressings are their low cost, defined structure, tunable properties, good mechanical properties, high chemical stability, and good antibacterial ability. The disadvantages are the complexity of synthetic polymer synthesis, single performance, and environmental unfriendliness. The monomer chemical structure of synthetic materials is shown in Table 3, and the contents of synthetic polymer and inorganic modified polyurethane materials discussed above are shown in Table 4.

### 3.3. Polyurethane Dressings Loaded with Other Bioactive Ingredients

Although polyurethane-related dressings are physiologically, mechanically, and economically superior to other dressing materials, they have poor healing capabilities and are considered passive wound dressings. Therefore, bioactive additives such as growth factors, biomolecules, or cells have been applied to polyurethane foam dressings to improve their healing qualities, and they are particularly suitable for the treatment of complex wounds (e.g., infected wounds, burn wounds, and diabetic wounds) that cannot be healed with conventional dressings [85].

#### 3.3.1. Multipotent Adult Progenitor Cells (MAPCs)

MAPCs are non-hematopoietic adherent cells derived from bone marrow, and preclinical evaluations of MAPCs have shown significant therapeutic benefits in improving tissue regeneration [86]. Advanced dressings for the delivery of MAPCs have been greatly developed in recent years [86,87], and a polyurethane-related dressing for the delivery of MAPCs is described below. A free radical-rich layer has been produced, by plasma immersion ion implantation (PIII) on medical polyurethane dressings that can attach biomolecules rapidly and covalently. The reactivity of polyurethane treated with PIII was used to immobilize the extracellular matrix protein tropoelastin, which could still maintain a functional conformation after sterilization with medical-grade ethylene oxide. MAPC adhesion and proliferation were promoted by tropoelastin-functionalized patches treated with PIII while preserving their cellular phenotype. In a topically applied MAPC patch, cells transfer to wounds on the skin, and untransferred MAPCs fill in the patch surface for subsequent cell transfer. Using such a wound patch, MAPCs and cytokines can be continuously delivered, enabling its use as a large-area dressing [88].

#### 3.3.2. Platelet Lysate

Chronic skin lesions are difficult to heal due to reduced levels and activity of endogenous growth factors. The platelet lysate, obtained by repeated freeze–thawing of platelet-enriched blood samples, is an easily attainable source of a wide range of growth factors and bioactive mediators involved in tissue repair [89]. A bilayer fibrin/polyether polyurethane scaffold loaded with platelet lysate is made by a combination of electrostatic spinning and spray phase-inversion. Enzyme-linked immunosorbent assays and fibroblast proliferation have been used to detect release and bioactivity of growth factors released from platelet cleavage scaffolds. Bilayer fibrin/polyether polyurethane scaffolds loaded with platelet lysate sustain the release of biologically active platelet-derived growth factors in vitro. An in vivo experiment revealed that the scaffold helped diabetic mice heal wounds more quickly. Histological results showed that platelet lysate and growth factor-loaded scaffold promoted collagen deposition and re-epithelialization in wounds of diabetic mice [90].

#### 3.3.3. Exosomes

Elevated oxidative stress, infection, reduced angiogenesis, and subsequent hypoxia are key factors in the non-healing of chronic diabetic wounds. The management and successful treatment of diabetic wounds remains a major therapeutic challenge, and the development of biological dressings with the ability to deliver oxygen, induce angiogenesis, and protect against oxidative stress and infection is important for the treatment of diabetic wounds [91]. Exosomes are cell-derived vesicles that carry large amounts of growth factors and tiny RNAs that maintain cellular homeostasis and regulate intercellular communication, including wound healing and angiogenesis [92]. OxOBand wound dressing is loaded with oxygen-releasing antioxidant exosomes, and it was developed specifically for promoting wound healing and skin regeneration in diabetic wounds. OxOBand is comprised of antioxidant polyurethane, a highly porous cryomaterial capable of sustained oxygen release, supplemented with adipose-derived stem cell (ADSC) exosomes. When applying ADSC exosomes and oxygen-releasing antioxidant scaffolds to a wound, fibroblasts and keratinocytes are able to attach, survive, migrate, and proliferate. In vivo results showed that OxOBand increases wound healing rate, re-epithelialization, and granulation tissue formation in diabetic rats. OxOBand treats diabetic wounds by promoting collagen remodeling, angiogenesis, and reducing oxidative stress [93,94].

#### 3.3.4. Adipose Stem Cell (ADSC)-Seeded Cryogel/Hydrogel Biomaterials

Adipose-derived stem cells have emerged as a promising tool for skin wound healing, but their therapeutic potential is largely dependent on the cell delivery system [95]. Hydrogels and cryogel biomaterials with antimicrobial properties are prepared from glycol chitosan and a novel biodegradable Schiff base cross-linking agent, difunctional polyurethane (DF-PU). Such a cryogel has a water absorption of ~2730 ± 400%, abundant macropores, 86.5 ± 1.6% formed by ice crystals, and a cell proliferation rate of ~240%, and hydrogels exhibit considerable antimicrobial activity and biodegradability. An adipose stem cell-seeded cryogel/hydrogel dressing was applied to the wounds of diabetic rats, and then acupuncture in Chinese medicine was performed to promote wound healing. The wound healing rate was as high as 90.34 ± 2.3%, with the wounds forming granulation tissues with sufficient micro-vessels and completing re-epithelialization within 8 days. By activating C5a and C3a, increasing the expression of cytokines TGF-β1 and SDF-1, and down-regulating proinflammatory cytokines IL-1β and TNF-α, the combination of acupuncture and stem cell-seeded cryogel/hydrogel biomaterials led to synergistic immunomodulation of the wound [96].

#### 3.3.5. L-Arginine (L-Arg)

L-Arginine is recognized as a conditionally essential amino acid for tissue growth in mature and juvenile mammals and has been used as a scavenger of reactive oxygen species in various species. In one study, polyurethane was used as a base polymer and blended with L-arginine to obtain desirable dressing properties such as better cell viability, cell attachment and proliferation, and enhanced antioxidant capacity of the dressing by blocking reactive oxygen species production [97]. A novel tissue adhesive (G-DLPU), constructed from L-Arg-based degradable polyurethane (DLPU) and GelMA, was prepared for wound care using the pro-angiogenic properties of L-Arg. After systematic characterization, G-DLPUs were found to have excellent shape-adaptive adhesion. In addition, the release of L-Arg during degradation and the production of NO were confirmed to contribute to wound healing. Biocompatibility was verified in in vivo experiments, and testing the hemostatic effect on damaged organs in a rat liver hemorrhage model showed that G-DLPUs reduced hepatic hemorrhage, with no significant inflammatory cells seen near the wound. Its therapeutic role in wound treatment was demonstrated in a mouse model of total skin defects, which showed that the hydrogel adhesive significantly improved the thickness of the neodermis and enhanced vascularization [98].

#### 3.3.6. LL37 Peptide

Antimicrobial peptides (AMPs) have therapeutic potential for treating bacteria and promoting skin regeneration. AMP LL37 peptide, an endogenous peptide in human skin, belongs to the antimicrobial family, and has antimicrobial, angiogenic, and immunomodulatory properties. LL37 peptide interacts with surface receptors, such as the epidermal growth factor receptor, on keratin-forming and endothelial cells. EGFRs and surface receptors such as formyl peptide receptor-like-1 (FPRL1) of keratinocytes and endothelial cells, respectively, mediate the migration of these cells and promote wound healing [99]. These studies evaluated the antimicrobial and pro-regenerative effects of LL37 peptides immobilized on a polyurethane-based wound dressing (PU-adhesive-LL37 dressing). The PU-adhesive-LL37 dressing killed Gram-negative and Gram-positive bacteria in human serum after 16 antimicrobial test cycles without inducing bacterial resistance. Importantly, re-epithelialization and wound healing were enhanced and wound macrophage infiltration was reduced in mice, with type 2 diabetes, treated with this new dressing compared to polyurethane-treated wounds of animals. Treatment of wounds of diabetic mice for 6 days with PU-adhesive-LL37 dressing resulted in a decrease in pro-inflammatory factor expression. In addition, the new dressing did not induce an acute inflammatory response compared with the control group. In summary, PU-adhesiative-LL37NP dressing may prevent bacterial infection, promote tissue contact for wound healing, and induce anti-inflammatory and re-epithelialization processes in diabetic wounds [100].

#### 3.3.7. Plasma Rich in Growth Factor (PRGF)

Growth factors such as PRGF serve as a rich source of active proteins that accelerate tissue regeneration. Animal experiments showed that PRGF-associated scaffolds contributed to skin wound healing and accelerated the formation of epidermal layers and skin appendages in rats [101]. In multilayered scaffolds created using PRGF from platelet-rich plasma, the outer layer is composed of polyurethane-cellulose acetate (PU-CA) fibers, while the inner layer is composed of PRGF-containing gelatin fibers. This approach, to prepare electrospun, biologically active scaffolds containing PRGF to induce cell proliferation and migration in vitro, is novel. Fluorescent images of fibroblast activity monitoring, using enhanced green fluorescent protein-labeled fibroblasts, showed that the migrating cell number on PRGF scaffolds was increased on day six. Real-time polymerase chain reaction analysis also revealed approximately 3-fold, 2-fold, and 2-fold increases in SGPL1, *DDR2*, and VEGF, respectively, on PRGF-containing scaffolds compared with cells migrating on PU-CA [102].

#### 3.3.8. Tri-Cell-Laden (Fibroblasts, Keratinocytes, and Endothelial Progenitor Cells) Hydrogels

Inadequate supply of donor skin limits the potential for treating severe wounds, and ex vivo engineered cell regeneration methods have been introduced as a viable alternative that promises to replace autologous skin grafting as the standard of care. The prevascularized mucosal cell sheet containing cultured keratinocytes, plasma fibrin, fibroblasts, and endothelial progenitor cells showed in vivo efficacy and tissue plasticity in cutaneous wounds by promoting accelerated healing [103]. A promising therapeutic strategy for treating inhomogeneous wounds is to fabricate customizable tissue-engineered skin. A planar/curved bioprintable hydrogel has been created that holds promise for the production of tissue-engineered skin. The dressing was evaluated in a rat irregular and chronic wound model. Gelatin and polyurethane are the main components of the hydrogel. There is excellent 3D printing ability and structural stability with polymer loaded with the three cell types. Treatment of circular wounds in normal and diabetic rats with planar-printed triple-cell-loaded hydrogels showed complete re-epithelialization and healing of the wound, and there was an abundance of new vessels and collagen after 4 weeks. Large, irregular skin wounds in rats treated with curvilinear-bioprinted, triple cell-loaded hydrogels showed wound repair was achieved after four weeks [104].

#### 3.3.9. Membranes Containing Mesoglycan and Lactoferrin

Thermoplastic polyurethane fiber membranes have been prepared using a uniaxial electrostatic spinning process. Fibers were then separately charged with two pharmacological agents, mesoglycan (MSG) and lactoferrin (LF), by supercritical CO_2_ impregnation. MSG and LF are uniformly distributed in a microscale structure. Angular contact analysis confirmed the fulfillment of MSG-loaded hydrophobic and LF-loaded hydrophilic membranes. The impregnation kinetics indicated that the maximum loadings of MSG and LT were 0.18 ± 0.20% and 0.07 ± 0.05%, respectively. Franz diffusion cells were used to simulate human skin contact in in vitro experiments. After 28 h, MSG release plateaued, whereas LF release plateaued after 15 h. Representing as human keratinocytes and fibroblasts, HaCaT and BJ cell lines have been evaluated for their compatibility with electrospun membranes in vitro [105].

In summary, the advantages of bioactive ingredients used as a source of dressings are significant therapeutic effects, good biocompatibility, and good antioxidant properties. The disadvantages are the high cost of bioactive ingredients, the scarcity of raw materials, the harsh storage conditions, and the difficulty of preservation. The contents of polyurethane dressings loaded with bioactive ingredients discussed above are shown in Table 5.

## 4. Discussion

### 4.1. Fabrication Techniques

Polyurethane polymers used as wound dressings are receiving more and more attention from scholars [8,106]. Polyurethane polymers have a relatively clear basic chemical structure that can be easily altered to add specific functional groups, and the materials are rarely associated with disease transmission and immunogenicity problems. Polyurethane polymers with adjustable soft and hard segments and modifiable chain extensions are widely used as materials for biological applications. The ratio or composition of hard and soft segments can be manipulated to alter the physicochemical properties of polyurethane [36,107,108]. As the clinical requirements for the functionality of the dressings increase, researchers in related fields have tried to modify the polyurethane with different ingredients or polymers in order to enhance multiple biological functions and physical properties [25]. By performing key biological functions, natural polymers sustain life and allow organisms to adapt to their environment. The worlds of synthetic and natural polymers are almost separate because synthetic polymers lack some specific biological functions; biochemical reactions caused by synthetic polymers can sometimes be uncontrolled and unwanted. Biologically active synthetic polymers with antimicrobial activity, among others, have been developed due to recent advances in synthetic polymerization techniques, such as antimicrobial activity, among others [109]. However, synthetic materials are less biodegradable and biocompatible, and the materials usually need to be combined with natural or other synthetic polymers to achieve the desired healing effect.

The preparation methods of dressings are various. In this review, electrospinning, molding, blending, composite, foaming, fiber bonding, hybridization, perfusion, solvent-free ring opening polymerization, bionic strategy, microemulsion method, one-step foaming, sol-gel, melt blowing, photopolymerization, and solvent-free phase separation are mainly involved. Due to the specific functional requirements of dressings used for wound healing, they are usually required to have specific performance such as antibacterial, biocompatible, adhesive, hydrophilic, antibacterial, mechanical properties, etc. Therefore, in order to meet the above performance requirements, the preparation methods of polyurethane dressings are usually not single, and most involve two or more manufacturing methods. The wound dressing is mainly based on polyurethane resin, which is blended or copolymerized with organic/inorganic materials to prepare composite materials with specific functions. In addition, reducing or loading methods can also be used to combine antibacterial, anti-inflammatory, and pro-healing therapeutic factors with polyurethane materials to prepare wound dressings suitable for special requirements.

Biomass materials have the advantages of wide sources, abundant raw materials, non-toxicity, and good biocompatibility, which will show huge development space in the field of wound dressings. The disadvantage is that its material properties are relatively single, so combining natural polymers and synthetic polymers can better meet the requirements of biomedical research, facilitate the utilization of their respective advantages, and achieve synergistic enhancement effects.

### 4.2. Biocompatibility Evaluations

Good biocompatibility is a prerequisite for the safe application of wound dressings in the clinic, so any dressing must undergo an adequate biocompatibility evaluation before it is applied in the clinic. Biocompatibility refers to the ability of a biomaterial to have an acceptable host response during the wound healing process, with the ability to be non-toxic, non-sensitizing, non-mutagenic, and non-carcinogenic [110]. Biomaterial biocompatibility studies to date can be categorized into animal, cellular, and molecular levels. Animal level evaluation can truly and comprehensively reflect the overall condition of the material after acting on the organism, but the cycle is long, expensive, complicated, not easy to control individual differences, and the results of animal experiments are not necessarily well applied to human beings. Cellular experiments have the advantages of simplicity, speed, sensitivity and economy, and they are easy to standardize and have good repeatability, but they cannot well simulate the complex physiological environment in the body. Changes at the cellular level and even at the overall level are caused by changes at the molecular level of the organism. Therefore, an in-depth study of the effects of biomaterials at the molecular level can reveal the mechanism of the interaction between the material and the organism. However, most of the biocompatibility investigations of polyurethane dressings mentioned above have been accomplished through cellular and animal experiments, and fewer molecular experiments have been carried out. Molecular experiments can be used to carry out basic research on biocompatibility, clarify the interaction relationship, elucidate the mechanism of dressing action, guide the research, development and application of new wound dressings, and lay the foundation for reducing the number of experimental animals and establishing new standards and methods for the safety evaluation of biological dressings.

### 4.3. Healing Evaluations

Skin wound repair is accomplished through a series of complex and highly coordinated processes [4], and is particularly susceptible to impairment by infection, inflammation, and oxidative stress, which prolong wound healing during the recovery process. In the face of the destruction of various factors, the rational design of intelligent and multifunctional wound dressings is imminent [111]. Elimination of bacterial infections is essential for better wound recovery. The inappropriate use and misuse of antibiotics in recent years has led to increased difficulty in treating wound infections, as bacteria reduce the penetration of antibiotics by forming biofilms or forcing antibiotics out of the body to reduce the concentration of antibiotics within the bacteria, weakening the antibiotic antimicrobial effect and creating resistance to the antibiotics [112,113]. Therefore, to overcome the hardship of antimicrobial resistance, the development of novel non-antibiotic strategies for difficult-to-treat drug-resistant bacterial infections has become a focus of much research, along with research on polyurethane-related antimicrobial materials.

The polyurethane-related dressings discussed in this article focus on some of the basic and necessary features of dressings, including antimicrobial properties, adhesion and hemostasis, anti-inflammation and anti-oxidation, substance delivery, and self-healing. However, less attention has been paid to the function of the skin after healing, especially the formation of skin appendages (such as hair follicles, sweat glands, and sebaceous glands), and scar, which are important structures affecting the function of the skin and have only been explored in a few articles [93,114,115]. Therefore, the development of polyurethane dressings that promote functional repair (protection, thermoregulation, modification) of the skin is the next priority.

Functional exploration of polyurethane dressings has primarily used rat, mouse, and rabbit models, with wound repair models from pigs being rarely used [116]. The main disadvantages of using rabbit or rodent models are the differences in skin physiology, healing patterns, and skin-attached hairs. In the field of skin healing, the pig model is considered to be a useful analog of human skin because it has many anatomical and physiological similarities and to human wound healing, and it is a better model for studying skin regeneration. Functional judgment of dressings in pigs would make the conclusions more clinically applicable [117].

Although polyurethane-related dressings have been observed to promote re-epithelialization, collagen deposition, and nerve repair in animal models, research has mainly focused on observation and analysis of experimental phenomena and lacks in-depth exploration of basic principles. Most of the studies did not assess whether the dressings could dynamically adapt to microenvironmental changes and healing at different stages displayed by chronic wounds, so as to realize the precise intervention of the dressings in order to accelerate the inflammation–proliferation–remodeling phase, and to realize the rapid and high-quality repair of chronic wounds. Overall, research on reforming clinical polyurethane dressings is still in the exploratory stage, but the preliminary findings display its potential clinical therapeutic value.

## 5. Conclusions and Future Perspectives

Nowadays, a variety of new polyurethane functional dressings have emerged as effective medical material candidates. As the clinical needs of skin wounds continue, the function of wound dressings needs to change from a single physical barrier or capability to the current multifunctional composite with a trend towards further intelligence [118]. Therefore, this paper presents a review of functional polyurethane dressings covered in the existing studies, which are mostly composed of composite materials. The composites include synthetic polymers, natural polymers, and other active ingredients, among others. In conclusion, the addition of polymers or active ingredients to polyurethanes can improve the functionality of the dressing, with natural polymers excelling in increasing the degradability and cytocompatibility properties of the dressing, synthetic polymers in antimicrobial and moisturizing properties, and other bioactive ingredients in promoting wound healing effects. In order to meet the actual application, they need to be compounded to utilize their advantages and realize cost savings.

Although these improved polyurethane-related dressings have been shown in studies to be multifunctional and intelligent, there are not many commercially available polyurethane dressings, and much progress has yet to be made in clinical applications. In the future research, with the development of composite material synthesis technology, the deepening of cognition, the innovation of treatment means, and the update of treatment guidelines, including electrospinning, 3D printing, scalp transplantation therapy, stem cell treatment, and genetic therapy, the biological materials for skin wound dressings have made great progress. Researchers can now address the significant need for new strategies for the treatment of chronic wounds, with the goal of breaking through the application bottleneck, and provide new design ideas and theoretical bases for treatment.

## Figures and Tables

**Figure 1 polymers-15-04301-f001:**
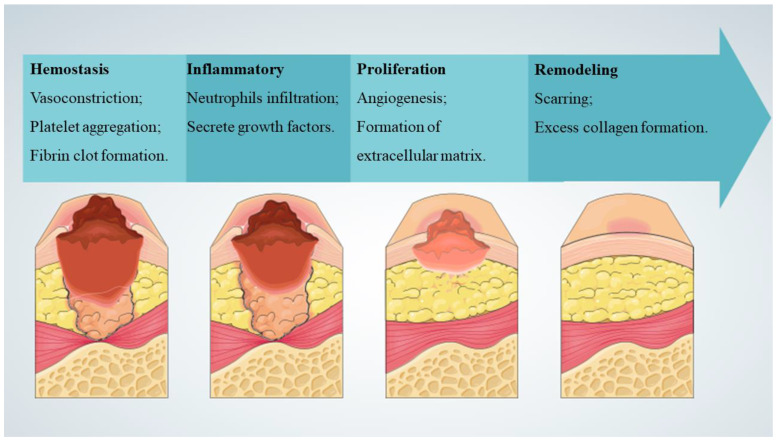
Typical stages of wound healing [4].

**Figure 2 polymers-15-04301-f002:**
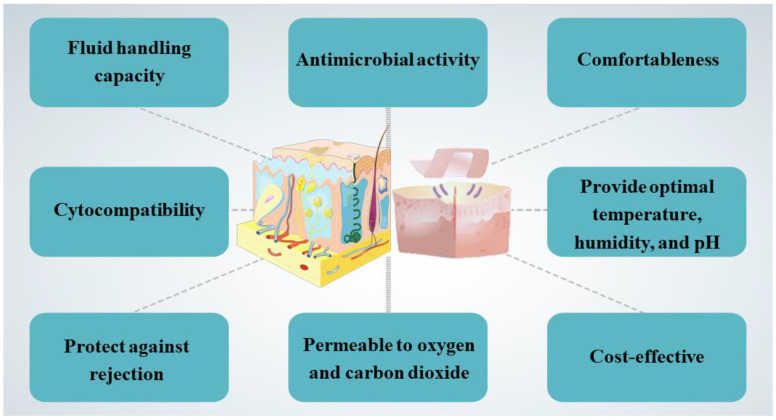
Characteristics possessed by an ideal dressing [8,9,10,11].

**Table 1 polymers-15-04301-t001:** Chemical structure of natural-based materials.

Collagen	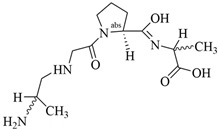	Tannic acid	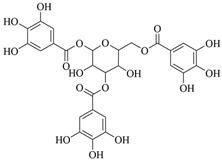
Chitosan	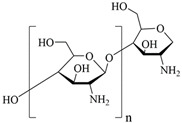
Hyaluronic acid	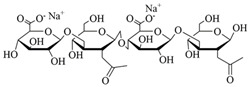	Thymol	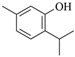
Lignin	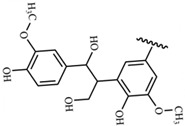

**Table 2 polymers-15-04301-t002:** Summary of methods and properties of natural-based polyurethane materials.

Natural Product	Process and Method	Research Models	Characterization
Collagen [31]	Microemulsion.	MTT Assay; antibacterial test; hemolysis test.	Enhances the mechanical properties and biocompatibility.
Chitosan [33]	Electrospinning.	MTT assay; trypan blue exclusion assay; DAPI staining.	Biocompatible.
Hyaluronic acid [35,36]	Coaxial electrospinning technique; crosslinked.	L929 cell viability; rat wound model; rat liver hemostasis model.	Biocompatible; non-toxic; promotes cell adhesion; shape-restoring ability; anti-inflammatory; enhances angiogenesis and regeneration of hair follicles.
Vegetable oil [38]	Polyaddition.	L929 cell viability; rat wound model.	Tensile strength; retention of moisture; cytocompatibility.
Tannic acid [40]	Polyaddition.	Diabetic mouse wound model.	Hemostatic; anti-moisture adhesion; anti-inflammatory; antioxidant.
Thymol [42]	Electrospinning.	Antimicrobial test.	Stretchable; breathable; moisturizing.
Lignin [45,46]	Dialysis; freeze-drying.	Oxidative stress model of LO2 cells; mouse burned skin model.	Antioxidant; promotes cell proliferation; non-cytotoxic; absorbency.
Peppermint extract [51]	Electrospinning.	MTT Assay; antibacterial test; diabetic rat wound model.	Anti-inflammatory; absorbent; promotes functional skin regeneration; antibacterial.
Gelatin [54]	3D printing; dialysis.	hMSCs culture in GelMA-PU cryogel.	High printing resolution; biocompatibility; adhesive, light transmittable; biodegradable.
Dextran [57,58]	Electrospinning.	In vitro degradation studies; vapor transmission rate analysis; blood compatibility evaluation; antibacterial activity.	Good hydrophilicity, water vapor permeability, adsorption rate and biodegradability, and promotes platelet adhesion and hemostasis.

**Table 3 polymers-15-04301-t003:** Monomer chemical structure of synthetic materials.

Acrylamide	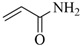	Ethylene glycol	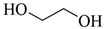
Caprolactone	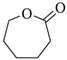	Vinyl alcohol	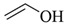
Lactic acid	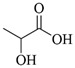	Cellulose acetate	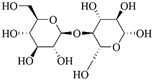

**Table 4 polymers-15-04301-t004:** Summary of methods and properties of synthetic polymer and inorganic modified polyurethane materials.

Synthetic Polymer and Inorganic Modified Polyurethane	Process and Method	Research Models	Characterization
Povidone-iodine [61,62]	Maceration.	Rat full-thickness skin defect model; prospective randomized case studies.	Promotes re-epithelialization, angiogenesis, collagen deposition, tissue invasion; absorbent.
Polyacrylamide [6]	One-pot method.	L929 fibroblast cytocompatibility assay; rabbit full-thickness skin defect model.	Superior stretch and ductility; adhesion; water absorption; moisture retention; antimicrobial; breathability.
Polycaprolactone [55]	Electrospinning.	L929 fibroblast cytocompatibility assay; rat wound model.	Hydrophilic; biodegradable; promotes collagen deposition; antimicrobial.
Polylactic Acid [71,72]	Polyaddition, electrospinning.	L929 fibroblast cytocompatibility assay.	Water absorption; biocompatibility.
Polyethylene glycol [74,75]	Self-foaming reactions.	Nondiabetic and diabetes mellitus rat wound models.	Absorbency and antiadhesion properties.
Polyvinyl alcohol [77]	SC/PL technique.	MTT assay; microbiology tests; cytotoxicity assay.	Antimicrobial; cytocompatible.
Tributylammonium alginate surface-modified cationic polyurethane [78]	Supramolecular ionic interactions.	Human dermal fibroblast model; infected and non-infected wounds in a rat full-thickness skin defect model.	Promotes fibroblast migration; hydrophilic; anti-inflammatory; promotes collagen deposition, angiogenesis; antibacterial.
Cellulose acetate/polyurethane nanofibrous mats containing reduced graphene oxide/silver nanocomposites and curcumin [79]	Improved Hummer method; hydrothermal method; electrospinning.	MTT assay using MEF cells; antibacterial test; C57 mouse wound model.	Moisturization; antimicrobial; promotes regeneration of the epidermal layer.
Nanosized copper-based metal-organic framework [80]	Crosslinking.	Antibacterial test; cytotoxicity assay mouse embryonic fibroblasts.	Selective antimicrobial capacity; cytocompatibility.
Silver [83]	Blending and light curing.	L929 fibroblast cytocompatibility assay and scratch assay; antibacterial test.	Antimicrobial; permeable to oxygen and carbon dioxide; tensile strength.

**Table 5 polymers-15-04301-t005:** Summary of methods and properties of polyurethane dressings loaded with bioactive ingredients.

Bioactive Ingredients	Process and Method	Research Models	Characterization
Multipotent adult progenitor cells [72]	Plasma immersion ion implantation; covalent attachment.	Human skin repair model.	Moisturizing; anti-hydrolytic; anti-inflammatory; modulates immune response; promote dermal and vascular regeneration; recruits other stem cells.
Platelet lysate [73]	A combination of electrospinning and spray.	Cell proliferation of mouse fibroblasts; diabetic mouse wound model.	Promotes capillary and collagen deposition; re-epithelialization; anti-inflammatory.
Exosomes [76,77]	Embedding.	Diabetic rat wound model; HaCaT, SH-SY5Y and NIH3T3 cell viability.	Enhances collagen deposition; increase neovascularization; reduces oxidative stress; promotes development of mature epithelial structures and hair follicle regeneration.
Adipose stem cell-seeded cryogel/hydrogel biomaterials [78]	Chemical synthesis.	Diabetic rat wound model; antibacterial testing.	Biodegradability; down-regulation of pro-inflammatory cytokines; angiogenesis; re-epithelialization.
L-Arginine [79]	Dialysis; freeze-drying.	Murine full-thickness skin defect model.	Shape-adaptive adhesion; biocompatibility; hemostasis; vascular regeneration; anti-inflammatory.
LL37 peptide [81]	Gum.	Antibacterial testing; cytotoxicity of human dermal fibroblasts; type II diabetic mouse wound model.	Antibacterial; anti-inflammatory; induces epithelialization.
Plasma rich in growth factor [83]	Electrospinning.	Human foreskin fibroblast cell viability.	Induction of fibroblast proliferation and migration.
Tri-cell-laden (fibroblasts, keratinocytes, endothelial progenitor cells) [84]	3D Planar-/Curvilinear-Bioprinting.	Rat fibroblast and keratinocyte viability; circular wound models in normal and diabetic rats.	Promotes vascularization, collagen regeneration; re-epithelialization.
Mesoglycan and lactoferrin [85]	Uniaxial electrospinning; supercritical impregnation.	Human immortalized keratinocytes and human immortalized fibroblast viability.	Biocompatibility; moisture control capability.

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
