# Peer review of "An Advanced Review: Polyurethane-Related Dressings for Skin Wound Repair"

_polymers, 2023, doi:10.3390/polym15214301_

Round 1

Reviewer 1 Report

Comments and Suggestions for Authors

This is a well review about the polyurethane (PU)-related dressings for skin wound. Please see the comments below.

- Several sentenses did not show the references.

- Reference number in parenthesis should be placed before any punctuation including comma, fullstop. 

- References in Table 1, 2 and 3 must be mentioned, This makes reviewers and readers difficult to follow the previous information.

- In 3.3.3 Exosome, the main idea sentence must be included. 

Comments on the Quality of English Language

Moderate editing of English language is required.

Reviewer 2 Report

Comments and Suggestions for Authors

In general, the writing of the paragraphs should be revised, there are errors that should be improved, the citation of references should be revised.

The section Categorization and Display Strategies seems more like a theoretical framework than a methodology. Based on these concepts, it should be explained what was done for categorization.

The PRISMA guidelines protocol was followed and the inclusion and exclusion criteria were made but it is not seen in the results, the prism diagram, the choice of articles, number of articles read, articles that were discarded, etc.

The words in vitro or in vivo, in italics 

Staphylococcus aureus in italics 

Escherichia coli in italics

In most articles good properties are mentioned, e.g.: good water absorption, good hydrophobicity, and good biocompatibility but this is according to what values or based on what can be considered good? the research is not compared with any reference value or any ideal value. 

In Table 1, Table 2 and Table 3 it would be better to place the references of each of the articles from which this information was summarized. 

More analysis of the topic should be added; there are only summaries of each research but no additional information is given. At the end of each category could be compared to see what are the advantages and disadvantages of each one. 

The conclusions and future perspectives that are presented do not show a connection with the results shown and an analysis of the different materials shown. It is seen as an introduction to the subject.

Taking into account that the first objective of the review is to systematically analyze the research results of polyurethane-related skin wound dressings, a systematic analysis is not performed since neither the systematic methodology used nor the results of the review are shown (number of total articles, decisions made, articles discarded after the review, PRISMA diagram, decision matrix, etc. of a systematic review), how many people carried out the review? How were the decisions made? And as for the results, only the research is presented but nothing is analyzed. 

And with respect to the second objective to provide strong support for further research in related fields by comprehensively sorting out and analyzing the un-solved problems, there is no mention of what the solutions could be, it is not shown based on the collection of information that was generated.

Reviewer 3 Report

Comments and Suggestions for Authors

The idea of the reviewed manuscript was to present the review article form, all issues related to the use of polyurethanes and its composites in the formation of dressings used in the regeneration and healing of therapeutically difficult skin wounds. Unfortunately, this work covers too little knowledge on this topic, does not contain appropriate tabular summaries - so useful for the reader, and practically does not contain illustrations, which are important in this type of study, which makes it difficult to read and does not fulfill  intended purpose of the article.

So far, according to the Scopus database about 90 review works on the issue of production and use of special polymeric composite dressings in the treatment of difficult-to-heal wounds have been published, including many medical systematic reviews. However, there is little work only focused on the use of polyurethanes and their composites for this purpose. The last work of this type is a systemic review that was published about a year ago in Polymers (Polymers 2022, 14, 2990. https://doi.org/10.3390/polym14152990).   Unfortunately, the authors treated this interesting topic in their work to much briefly and superficially, as evidenced by the small selection of cited literature and the relatively small scope of topics discussed. In this way, they presented a poor work not suitable for publication in Polymers. Below are the most important allegations that contributed to my decision.

- the entire work was designed very poorly - the chapters and their order, which causes the information to be presented chaotically and without a sensible plan;

- lack of a completely original approach to the collected literature data, lack of any division and order of presentation of these data according to a sensible key, information is incomplete and provided in a very chaotic way;

- data on the type and chemical structure of polyurethanes used, methods of their preparation, compositions of composites, etc. are practically omitted in the descriptions;

-lack of description of the results of preclinical and clinical studies using the dressings that are the subject of the manuscript;

-lack of description of commercially available polyurethane dressing systems, their application, and treatment effects;

- complete lack of illustrations, diagrams, and diagrams necessary in this type of work;

Comments on the Quality of English Language

I didn't notice any major linguistic or grammatical errors. The text is clear and linguistically understandable.

Reviewer 4 Report

Comments and Suggestions for Authors

This review systematically describes the changes in physicochemical and biological properties brought about by the incorporation of different polymers into polyurethane dressings, and describes their applications in wound repair and regeneration.

The manuscript is in principle interesting, however in the introduction limited space is dedicated to the cellular and molecular mechanisms involved in wound healing. This concept is exasperated in the few lines dedicated to  chronic wounds.

Besides is necessary to add some specific references of these research field. These references (at least 4 or 5) have to be recent (2022-2023)

Comments on the Quality of English Language

 Moderate editing of English language required

Round 2

Reviewer 2 Report

Comments and Suggestions for Authors

The authors made the requested modifications and the review has a greater contribution to the field.

Author Response

Thank you for your careful review.

Reviewer 3 Report

Comments and Suggestions for Authors

The current version of the manuscript is certainly a version of the greater value and usefulness to the reader. As I wrote earlier, the advantage of this article is the selection of a description of one group of dressings, dressings based on polyurethane products. Unfortunately, the main topic related to the title of the article is little exposed in the text of the manuscript. The authors should supplement the article with more detailed descriptions, preferably in the form of a separate chapter in which the following issues will be presented, supported by literature examples;

- why are composite dressings with a polyurethane matrix chosen, what are the reasons for this, what are the advantages of such solutions and what are the disadvantages compared to the most popular dressings of other types;

- how the composite polyurethane dressings described in the article are formed, how are structure, more about the manufacturing techniques used, the impact of processing on the properties of dressings, possibilities of mass production, and commercially available products of this type.

After making these additions, the work can be published.

Comments on the Quality of English Language

It was written in a clear language, without major spelling, syntax or dictionary errors.

Reviewer 4 Report

Comments and Suggestions for Authors

The authors have answered correctly to my questions.

Comments on the Quality of English Language

 Moderate editing of English language required

Author Response

Thank you for your careful review. We have further improved the English language of the manuscript, you can refer to the part of the manuscript marked in red.